# Dust Lidar Ratios Retrieved from the CALIOP Measurements Using the MODIS AOD as a Constraint

**Man-Hae Kim [1], Sang-Woo Kim [1,\*]** 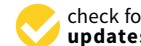 **and Ali H. Omar [2]**

1   School of Earth and Environmental Sciences, Seoul National University, Seoul 08826, Korea; manhae@snu.ac.kr
2   NASA Langley Research Center, Hampton, VA 23666, USA; ali.h.omar@nasa.gov
\*   Correspondence: sangwookim@snu.ac.kr

**Abstract:** Lidar ratio for dust aerosols is retrieved from a synergetic use of the Cloud-Aerosol Lidar with Orthogonal Polarization (CALIOP) Total Attenuated Backscatter coefficients and the Moderate Resolution Imaging Spectroradiometer (MODIS) Aerosol Optical Depths (AODs) for 5 years from 2007 to 2011. MODIS AODs from the Dark Target (DT) algorithm over ocean and from the Deep Blue (DB) algorithm over land are used as a constraint for the retrieval. The dust lidar ratio is retrieved larger over land (46.6 ± 36.3 sr) than ocean (39.5 ± 16.8 sr) and shows distinct regional variation. Lidar ratio for Saharan dust (49.5 ± 36.8 sr) is larger than Arabian dust (42.5 ± 26.2 sr). Lidar ratios for dust aerosols transported to Mediterranean Sea (44.4 ± 15.9 sr), Mid Atlantic (40.3 ± 12.4 sr), and Arabian Sea (37.5 ± 12.1 sr) show lower values relative to their source regions. Retrieved dust lidar ratios for Taklamakan and Gobi Deserts region (35.0 ± 31.1 sr) and Australia (35.4 ± 34.4 sr) are slightly lower than the above-mentioned regions. AOD comparison between CALIOP and MODIS shows that the CALIOP AOD is biased low. When including clear air AOD for CALIOP, however, AODs from two sensors become more comparable.

**Keywords:** lidar ratio; dust; aerosol optical depth (AOD); CALIOP; MODIS

## 1. Introduction

The Cloud-Aerosol Lidar with Orthogonal Polarization (CALIOP) onboard the Cloud-Aerosol Lidar and Infrared Pathfinder Satellite Observation (CALIPSO) satellite is a space-borne lidar system measuring vertical profiles of aerosols and clouds over the globe since 2006 [1]. The CALIOP level 2 aerosol products provide vertically resolved aerosol extinction as well as aerosol types [2]. While the vertical resolution of aerosol properties provided by the CALIPSO is invaluable, many studies report lower aerosol optical depths (AODs) for CALIOP compared to other space-borne, airborne, or ground-based measurements [3–9]. There are two main reasons for the lower CALIOP AOD; inaccurate lidar ratio (i.e., extinction-to-backscatter ratio) including misclassification of aerosol types [8,10–12] and undetected or missed aerosol layers [9,13–15].

In November 2016, a new version (version 4) of the CALIPSO data products was released. Because of several improvements in version 4 [16–21] CALIOP AOD has increased by 40% (52%) over version 3 for daytime (nighttime). The comparison of CALIOP AOD with Aerosol Robotic Network (AERONET) and the Moderate Resolution Imaging Spectroradiometer (MODIS) over ocean shows that the AOD differences have decreased [18]. The lidar ratio, one of the most important parameter in aerosol extinction retrieval, is estimated to have uncertainty (defined as standard deviation of lidar ratios) at 30–50% in version 3. These have improved in version 4 for some aerosol subtypes such as clean

marine, dust, and smoke. Dust lidar ratio (±uncertainty) has changed from 40 ± 20 sr in version 3 to 44 ± 9 sr in version 4 with an improvement in uncertainty from 50% to 20%. However, the dust lidar ratio has large regional variability ranging roughly from 35 to 60 sr [11,22,23], which can lead to uncertainty in CALIOP aerosol retrieval for dust since same lidar ratio is used for all dust at any region over the globe. The regional variability of dust lidar ratio can be strongly affected by the mineralogical composition and refractive index of dust particles [11]. Dubovik et al. [24] also reported that the dust lidar ratio is sensitive to the particle non-sphericity because of the large difference of the scattering phase function between spherical and non-spherical particles.

One way to avoid the errors induced by inaccurate lidar ratio for the extinction retrieval from elastic lidar measurements is using independently measured AODs as a constraint. There are many studies using AOD constrained retrieval from ground-based and airborne lidar measurements [23,25–27]. In this study, dust lidar ratios are determined from the CALIOP (version 4) measurements by constraining the retrieval with MODIS (collection 6) AOD. As parts of A-train constellation [28], a synergetic use of CALIOP and MODIS (onboard Aqua satellite) provides many opportunities for collocated measurements globally since both instruments are in the A-train constellation separated by only a few minutes.

There are seven aerosol types in the CALIOP aerosol products [18]. However, we focused only on dust because dust aerosols are globally distributed on both land and ocean and have a broad range of lidar ratios. Therefore, by using AOD constrained retrieval, dust lidar ratios are determined globally and regional variability of dust lidar ratio are investigated in this study. Aerosol type classifications from the CALIOP level 2 aerosol product are used to identify and select dust aerosol layers. Since dust has the unique characteristic of large depolarization ratio, aerosol type classification in the CALIOP algorithm is relatively more accurate than other types of aerosols [12]. A detailed methodology for the AOD constrained retrieval is described in Section 3. The retrieved dust lidar ratios are stratified by region and compared to previous studies in Section 4.

## 2. Data

### 2.1. CALIOP

The CALIOP level 1B profile, level 2 aerosol profile, and level 2 merged aerosol and cloud layer products (version 4) are used in this study. The total attenuated backscatter (TAB) at 532 nm is used to retrieve aerosol extinction coefficients and lidar ratios. TAB is the calibrated, range-scaled, energy- and gain-normalized, baseline subtracted lidar return signal provided by the CALIOP level 1B profile product [16,29]. The fundamental resolution of CALIOP is 30 m vertical and 1/3 km horizontal at 532 nm [30]. Because data above 8.2 km are averaged vertically and horizontally before downlinking from the satellite, the vertical resolution for the released level 1B profile (532 nm) is 60 m for 8.2–20.2 km, 180 m for 20.2–30.1 km, and 300 m for 30.1–40 km.

The level 2 aerosol profile product contains profiles of aerosol extinction coefficient and aerosol subtypes retrieved from the level 1 product [2,18,19,31]. The spatial resolution for the level 2 profile product is 5 km in the horizontal and 60 m in the vertical dimension from the surface to 20.2 km (180 m for 20.2–30.1 km). The level 2 aerosol profile product is used in this study to determine dust layers. Since the level 2 profiles have coarse vertical resolution, the level 1B TAB profiles are vertically averaged in 60-m bins from the surface to 8.2 km before retrieval. The level 2 merged aerosol and cloud layer product (333 m) is used to exclude level 1B profiles which contain clouds.

### 2.2. MODIS

The MODIS provides AODs with horizontal resolution of 10 km × 10 km near nadir over land and ocean. The MODIS level 2 aerosol product (collection 6) from Aqua satellite (MYD04_L2) is used in this study. There are two different algorithms for the MODIS AOD retrieval; Dark Target (DT) for both land and ocean and Deep Blue (DB) over bright land surface [32,33]. Since the DT algorithm rarely

provides AODs over the desert areas, we used DT AODs over ocean and DB AODs over land. DT AODs are provided at seven wavelengths of 0.47, 0.55, 0.66, 0.86, 1.24, 1.63, and 2.13 μm over the ocean whereas DB AODs are at three wavelengths of 0.412, 0.47, and 0.66 μm over land. Additionally, DB product provides corrected AODs at 0.55 μm. AODs at 550 nm are used for both DT and DB in this study. The AOD difference between the 532 nm channel of CALIOP and the 550 nm channel of MODIS because of wavelength differences is assumed and can be shown to be negligible. Cloud mask and quality assurance (QA) data are also used for data screening as described below.

### 2.3. Collocation and Data Sellection

The CALIOP level 1 and level 2 products closer than 10 km from the center of MODIS level 2 data pixel are selected as the collocated data pairs as described in Kim et al. [15]. The distance between CALIOP ground track and the center of MODIS pixel should be less than 5 km for the collocation. The horizontal resolution for the CALIOP level 1 and 2 profile products are 333 m and 5 km, respectively, whereas the MODIS level 2 aerosol product provides AOD for 10 km × 10 km near nadir. Thus, each collocated data pair consists of one MODIS level 2 data pixel, 50–60 profiles of the CALIOP level 1B data and 3–4 profiles of the CALIOP level 2 data. The CALIOP level 1 and 2 profiles for each collocated data pair are averaged for the retrieval.

Aerosol extinction coefficients and lidar ratios are retrieved from the CALIOP profiles which contain only dust layers in this study. Dust layers are selected using the aerosol subtype classification provided by the CALIOP level 2 aerosol profile product. Since dusty marine and polluted dust are frequently found within the dust layers, dust layers consisting of up to 10% dusty marine or polluted dust are considered dust layers in this study.

Clouds are screened out in both CALIOP and MODIS data. The collocated data pairs which contain cloud optical depth higher than zero from the CALIOP level 2 profile product or cloud fraction higher than 30% from the MODIS level 2 aerosol product are excluded. The CALIOP level 2 333-m merged layer product is used to remove small clouds which are not reported by the CALIOP level 2 profile product (horizontal resolution of 5 km). The cloud aerosol discrimination (CAD) score is also used to remove clouds that are misclassified as aerosol [20]. Aerosol layers with CAD scores higher than −30 are not used in this study.

We use the QA flags of both data pairs to filter the data. MODIS AODs with QA higher than 1 (marginal) for DT over ocean and 3 (very good) for DB over land are used for the retrieval. The totally attenuated CALIOP level 1B profiles are excluded. The TAB profiles which do not reach the surface are considered as totally attenuated and screened out. Quality assurance for the CALIOP level 2 product is not necessary because the CALIOP level 2 aerosol extinction profiles are not used. For the comparison of AOD between CALIOP and MODIS in Section 5, however, extinction quality control (QC) flags of 0, 1, 16, and 18 from the CALIOP level 2 aerosol profile product are selected in this study [34].

The retrieved lidar ratios in this study are strongly affected by the accuracy of the MODIS AOD. The expected errors of MODIS AOD (collection 6) are +(0.04 + 10%), −(0.02 + 10%) for DT over ocean and ±(0.03 + 20%) for DB over land for typical air mass factor of 2.8 [32,33]. CALIOP, on the other hand, has larger expected error of ±(0.05 + 40%) in version 3 [8]. Since the uncertainty of dust lidar ratio has improved from 50 to 20% in version 4 [18], the expected error of CALIOP AOD for dust is smaller. The accuracy of MODIS AOD is comparable or better than CALIOP AOD and therefore reliable for use as the constraint for the AOD retrieval from CALIOP TABs. Generally, the relative uncertainty of MODIS AOD increases as the AOD decreases, thus low MODIS AOD can affect the uncertainty of the retrieved lidar ratio. However, only 9% of all the AOD data used in this study are lower than 0.05 and the influence of large relative error for low MODIS AODs is not significant except for Australia. Therefore, we used all available MODIS AOD data which satisfied the above coincidence criteria, except for data pairs with large AOD difference between CALIOP and MODIS as discussed in Section 4.

## 3. AOD Constrained Retrieval

The aerosol extinction retrieval method using the MODIS AOD as a constraint is described in Figure 1, which is similar to the method used in Kim et al. [15]. Since CALIOP retrieves only for the detected layers whereas MODIS provides AOD for the whole column of the atmosphere, clear air (defined as a region where no aerosols are detected above the detected dust layers) can be one of the main reasons for AOD discrepancy between CALIOP and MODIS. To minimize the errors caused by undetected or missed aerosol layers, the retrieval is performed from 30 km above the mean sea level to the surface. The lidar ratio used for clear air is a constant value of 30 sr following the work of Kim et al. [15]. In that work, they retrieved lidar ratios for undetected layers from the CALIOP measurements and found global mean (median) value of 32.62 (28.75) sr. Aerosol extinction coefficient (and lidar ratio) is retrieved from the renormalized TAB ($\beta'_N$) which is defined as (Equation (1))

$$\beta'_N(z) = \frac{\beta'(z)}{T_m^2(r_0, r_N)T_o^2(r_0, r_N)} = \left[\beta_m(z) + \beta_p(z)\right]T_m^2(r_N, z)T_p^2(r_N, z), \tag{1}$$

where $\beta'(z)$ is the TAB at the altitude of $z$ provided by the CALIOP level 1B profile product; $T^2(r_1, r_2)$ is the two-way transmittance from $r_1$ to $r_2$; $\beta(z)$ is the backscatter coefficient at the altitude of $z$; $r_0$ and $r_N$ are the calibration altitude (36–39 km) and renormalization altitude (30 km); and subscripts $m$, $o$, and $p$ represent air molecule, ozone, and aerosol, respectively. $\beta_m(z)$ and $T_m^2(r_N, z)$ in Equation (1) are calculated from the vertical profile of the molecular number density provided by the CALIOP level 1B product. The two-way transmittance of aerosols, $T_p^2(r_N, z)$, in Equation (1) can be expressed by (Equation (2))

$$T_p^2(r_N, z) = exp\left[-2\int_z^{r_N} \sigma_p(z')dz'\right] = exp\left[-2S_p\int_z^{r_N} \beta_p(z')dz'\right], \tag{2}$$

where $\sigma_p(z)$ is the aerosol extinction coefficient and $S_p$ is the lidar ratio which is defined as $\sigma_p(z)/\beta_p(z)$. Although $S_p$ is a function of altitude, it is assumed constant within the detected dust layer in this study as shown in Equation (2). The dust lidar ratio of 44 sr from the CALIOP version 4 aerosol retrieval algorithm [18] is used as an initial value for the detected dust layers in this study.

The profile of aerosol extinction coefficient is retrieved using an iterative method as described in Young and Vaughan [31]. AOD ($\tau_{CAL}$) is obtained from vertical integration of the retrieved aerosol extinction and compared with MODIS AOD ($\tau_{MOD}$; DT over ocean and DB over land). Then the lidar ratio for dust layer is modified until the AOD difference between the two instruments is less than 1%. Sometimes, however, the lidar ratio does not converge and the retrieval fails. This can happen when the AOD difference between CALIOP and MODIS is large, or the dust layer is shallow (physically thin). In about 31% and 40% of the collocated data pairs (cloud screened and quality assured) for ocean and land, respectively, the retrieval fails to converge. These cases are excluded from the analyses.

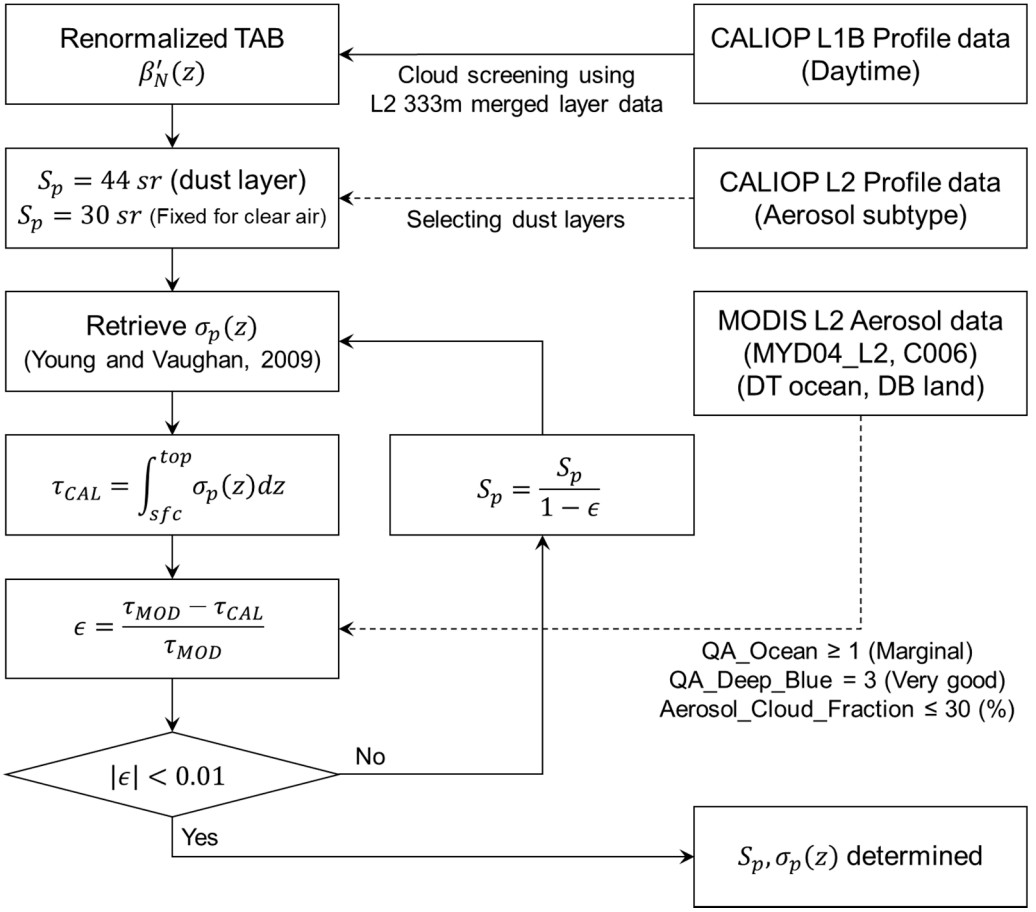

**Figure 1.** A flowchart for the aerosol extinction retrieval for dust layers using Moderate Resolution Imaging Spectroradiometer (MODIS) Aerosol Optical Depths (AOD) as a constraint, modified from Figure 1 of Kim et al. [15].

An example of AOD constrained retrieval over the Arabian sea in 18 October 2009 is shown in Figure 2. Based on the total attenuated backscatter at 532 nm (Figure 2a), the CALIOP level 2 aerosol algorithm determines aerosol layers and subtypes (Figure 2b) and retrieves aerosol extinction for the detected aerosol layers (Figure 2c). Figure 2d is aerosol extinction retrieved in this study using the AOD constrained method described in Figure 1. Note that only dust layers are analyzed in Sections 4 and 5 but Figure 2d shows all retrieved layers regardless of the aerosol subtype. The structures of aerosol layers are almost identical between Figure 2c,d except for some non-zero values above the aerosol layer in Figure 2d. This is because this study retrieves the aerosol extinction for the whole profile from 30 km to the surface whereas the CALIOP level 2 aerosol algorithm retrieves extinction only for the detected aerosol layers. Missing profiles in Figure 2d are due to lack of MODIS AOD, cloud screening, or retrieval failure, as described above. There is a clear air region (no aerosol is detected) inside the dust layer in Figure 2b,c at around altitude of 1 km and longitude of 60.86° which is not clearly shown in Figure 2d. Since CALIOP measures downward from space, detection for the layer top altitudes is relatively correct. Layer base altitude or layer boundaries for multiple layers, however, can be inaccurate. In this study, for the purpose of the retrieval, all dust layers are considered as a single layer from top of the highest layer to the surface. We ignore clear (undetected) areas below elevated layers or between multiple layers. Figure 2e shows AODs from MODIS and CALIOP level 2 products and retrieved lidar ratios.

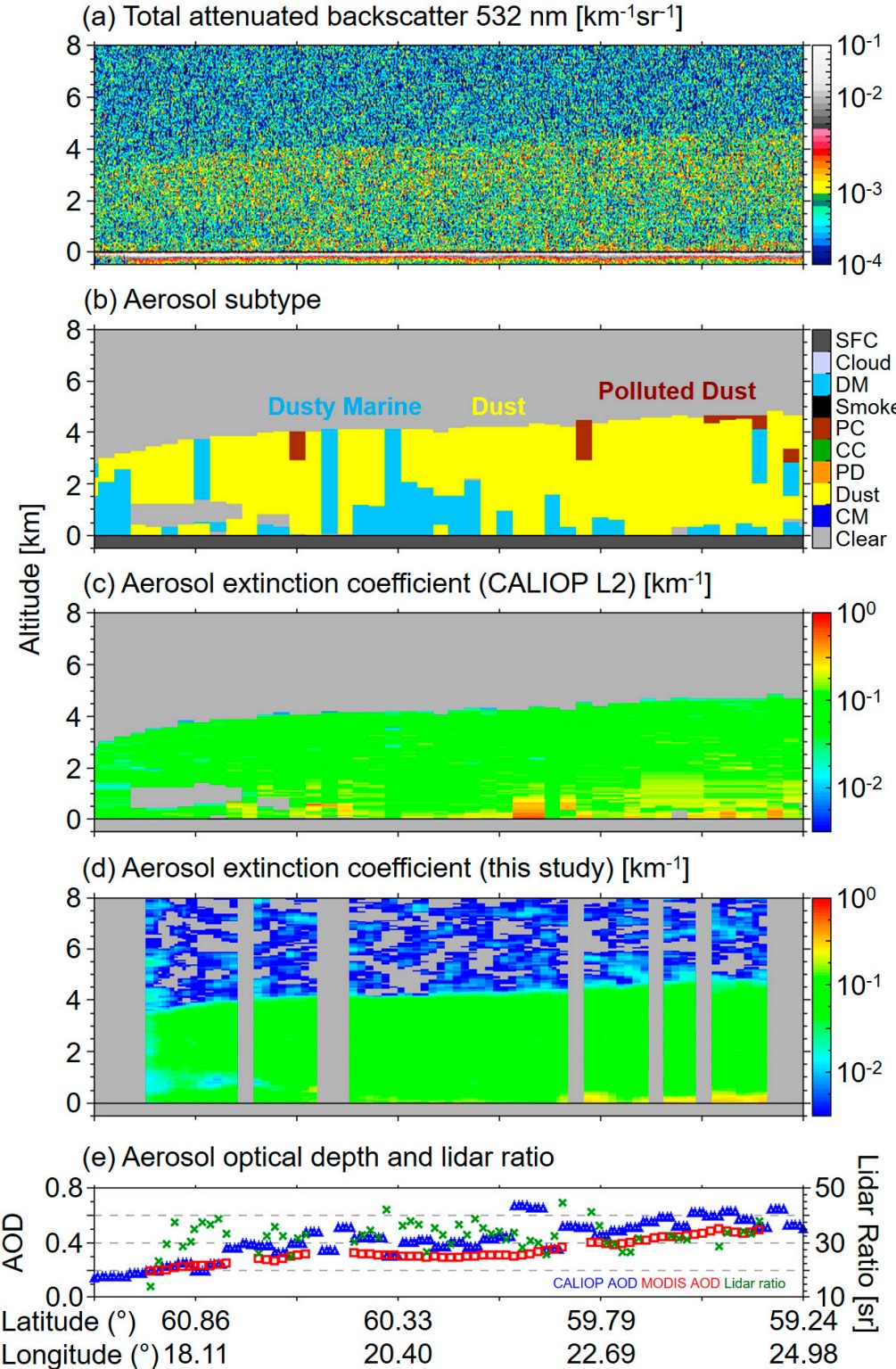

**Figure 2.** Cloud-Aerosol Lidar with Orthogonal Polarization (CALIOP) measurements of (**a**) total attenuated backscatter at 532 nm, (**b**) aerosol subtypes, (**c**) aerosol extinction from the CALIOP level 2 profile product, (**d**) aerosol extinction retrieved in this study, (**e**) aerosol optical depths from CALIOP level 2 product (blue triangles) and MODIS level 2 aerosol product (red squares) and lidar ratios retrieved in this study (green x-marks) over the Arabian sea in 18 October 2009 from 09:32:59 UTC to 09:34:50 UTC.

## 4. Results and Discussion

Lidar ratios are retrieved using AOD constrained method described in Section 3 for dust layers for the years 2007 to 2011. Lidar ratios retrieved in this study are strongly affected by AOD difference between CALIOP and MODIS. In other words, the retrieved lidar ratio increases when CALIOP AOD is lower, but decreases when MODIS AOD is lower to compensate the AOD difference. Figure 3 is the distribution of AOD difference between MODIS and CALIOP for dust layers used in this study. Mean, standard deviation (SD), and median values are shown together in the figures. Note that the CALIOP AOD in Figure 3 is from the CALIOP level 2 aerosol product that is not the retrieved AOD in this study. Because the retrieval is constrained by the MODIS AOD, the retrieved AOD (including clear air AOD) is identical to the MODIS AOD and not used for the comparison here.

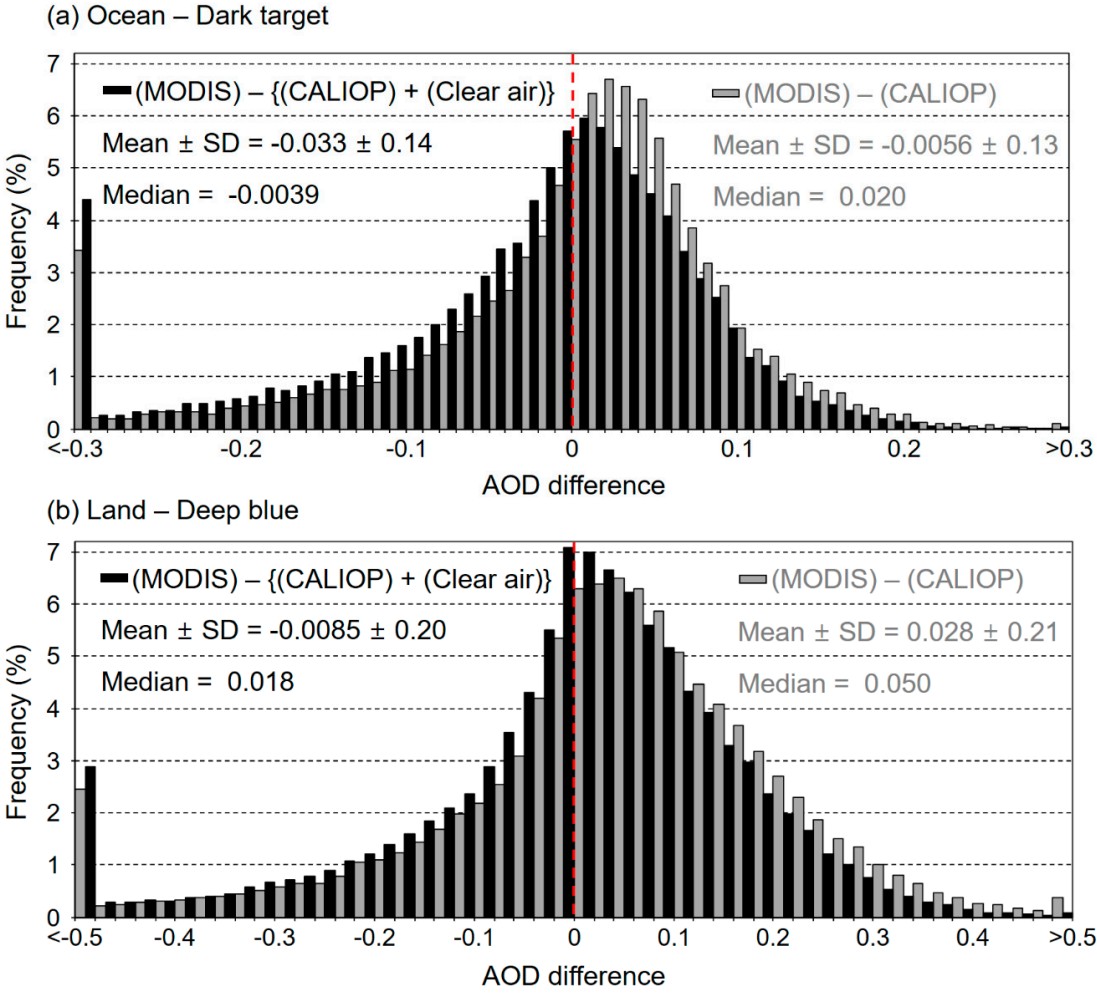

**Figure 3.** Frequency distribution of AOD difference between CALIOP and MODIS (gray bars) for dust layers used in this study from 2007 to 2011 (**a**) over ocean with MODIS DT AOD and (**b**) over land with MODIS DB AOD. Black bars represent the AOD difference including clear air AOD for CALIOP AOD. CALIOP AOD is from the CALIOP level 2 aerosol product and clear air AOD is retrieved in this study.

Gray bars in Figure 3 are the AOD difference between the two sensors. The center of distributions (or median) shows that the CALIOP AOD is biased low compared to the MODIS AOD for both ocean and land. When including clear air AOD for the CALIOP AOD (black bars in Figure 3), the center of the distribution moves to zero. This implies that the clear air AOD is responsible for the lower bias in the CALIOP AOD. If the CALIOP AOD is retrieved for the whole column of the atmosphere, the CALIOP AOD is closer to the MODIS AOD. As shown in Figure 3, AOD differences are sometimes

very large which can result in unphysical lidar ratios. Although QA/QC data are selected for both CALIOP and MODIS in Figure 3, large AOD difference implies that AOD from either or both of sensors may still have uncertainty. To remove data pairs with large AOD difference from the analysis, data pairs with AOD difference smaller than 1SD (−0.14 < AOD difference < 0.14 for ocean, and −0.20 < AOD difference < 0.20 for land, including clear air for CALIOP) are used for further analysis. Since the retrieved lidar ratios can be very sensitive with small changes in the AOD difference for low AODs, data screening with same threshold (1SD) is applied for the relative AOD difference. With this data screening, data pairs with large AOD differences are rejected from the analysis by about 25%.

Figure 4 shows the frequency distribution of the retrieved dust lidar ratios over ocean and land after discarding data pairs with large AOD difference as described above. Note that the *x*-axis is a logarithm scale and the lidar ratios show a log-normal distribution centered (mode) at ~41 sr for ocean and ~43 sr for land. The mean (and SD) lidar ratios are 39.5 ± 16.8 sr and 46.6 ± 36.3 sr over ocean and over land. This is reliable because the lidar ratio of 44 sr for dust used in the CALIOP level 2 aerosol retrieval algorithm lies between these two values [18]. However, median values of 38.1 sr over ocean and 39.2 sr over land are smaller than mean values. Few values smaller than 5 sr and larger than 200 sr (0.8% over ocean and 2.8% over land) are omitted in Figure 4. However, we accept all values of the retrieved lidar ratio for mean and median, except for values less than 1 which is physically impossible, to avoid bias induced by arbitrary criteria for the retrieved lidar ratios.

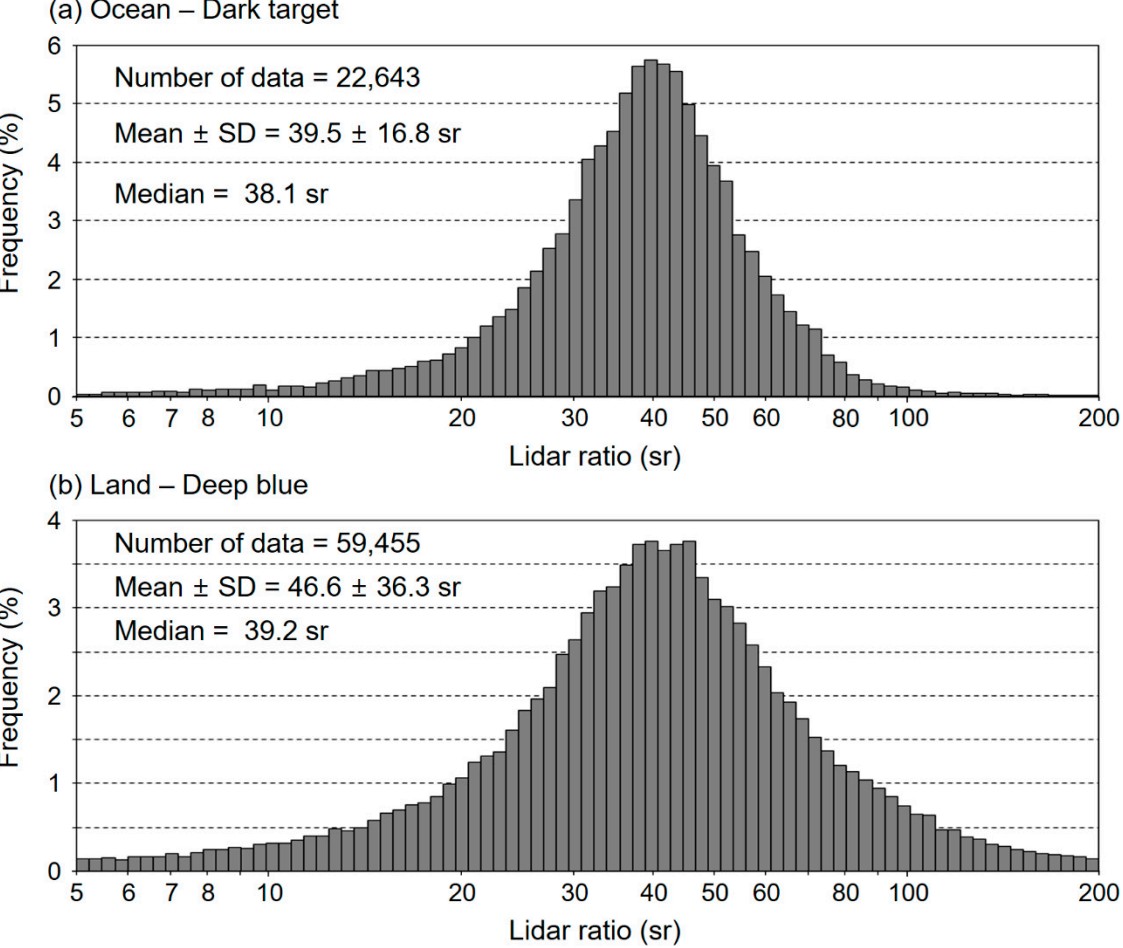

**Figure 4.** Frequency distribution of the retrieved dust lidar ratios (**a**) over ocean using MODIS DT AOD as a constraint and (**b**) over land using MODIS DB AOD as a constraint from 2007 to 2011. Note that the *x*-scale is logarithmic. Tails for both ends (lower than 5 sr and higher than 200 sr) are omitted in the figure but included for the statistics.

Global distributions of the lidar ratios over ocean and land are shown in Figure 5. We focus on several major dust source and outflow regions as shown with white dashed boxes in Figure 5; Mid-Atlantic (Rg1), Mediterranean Sea (Rg2), Arabian sea (Rg3), East Asian outflow region (Rg4) over ocean and Saharan Desert (Rg5), Arabian Desert (Rg6), East Asia including Taklamakan and Gobi Deserts (Rg7), Australia (Rg8) over land. There are some extreme values for the rest of these regions mainly because of insufficient number of data, which is not discussed in this study. Box-whisker plots of the dust lidar ratios for ocean and land as well as each region described above are shown in Figure 6. The numbers of data are shown on the top of the figure. The retrieved dust lidar ratio shows distinct regional variation; larger in Rg2 whereas relatively lower in Rg1 and Rg3 over ocean. Over land, dust lidar ratio is large in Rg5 but small in Rg7 and Rg8.

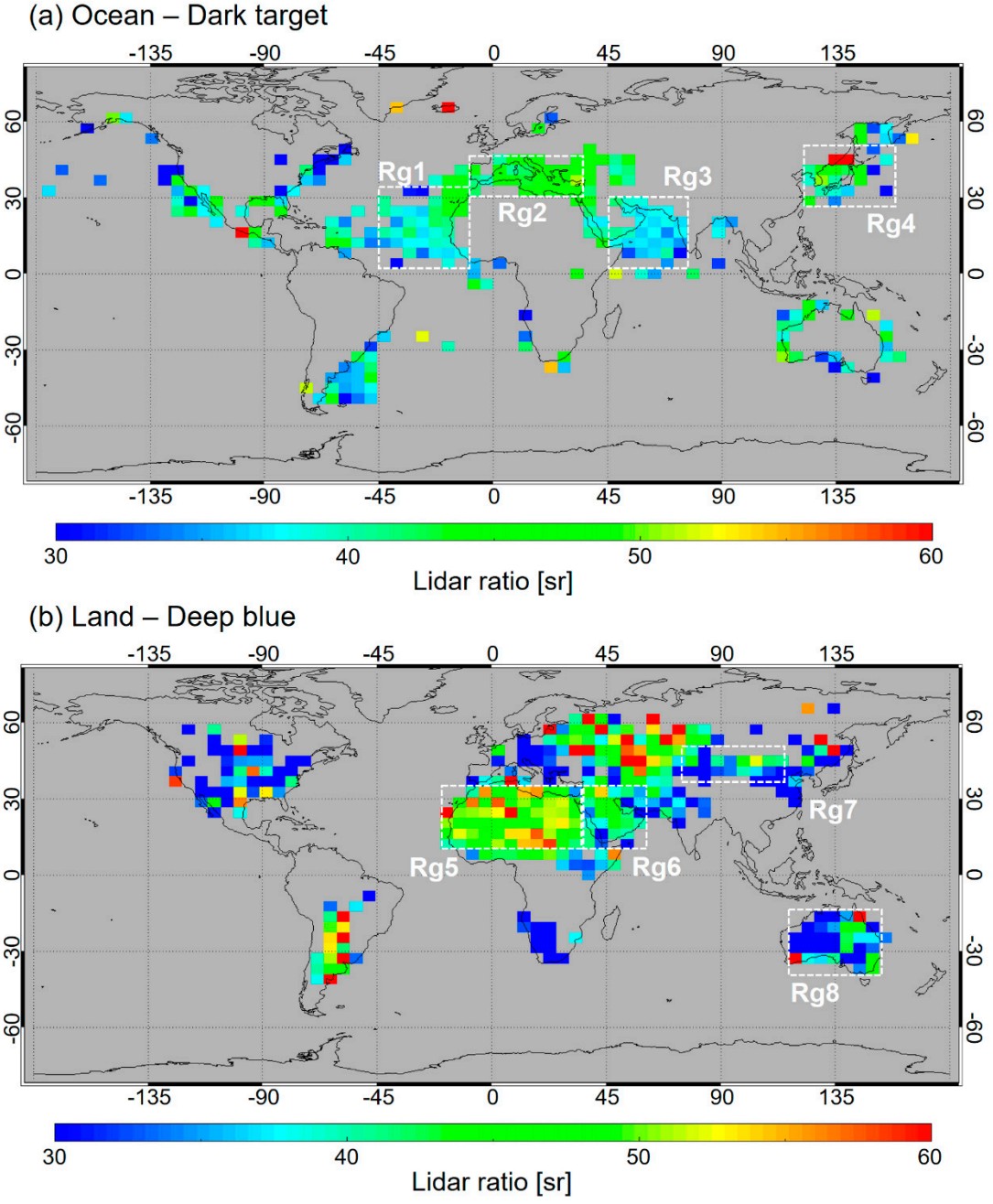

**Figure 5.** The global distribution of the retrieved lidar ratio (**a**) over the ocean using MODIS DT AOD as a constraint and (**b**) over land using MODIS DB AOD as a constraint from 2007 to 2011.

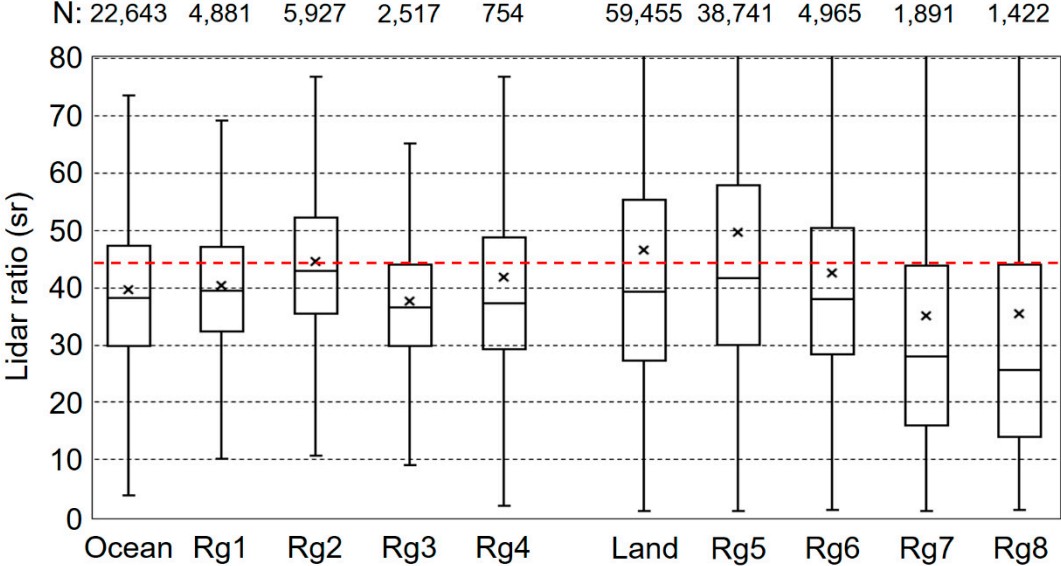

**Figure 6.** Box-whisker plots of retrieved lidar ratio over ocean, land and for regions shown in Figure 5. Boxes are for 50% percentile, bars and x-marks in each box are median and mean values, respectively. Red dashed line represents 44 sr that is used in the CALIOP version 4 aerosol retrieval algorithm. The number of data (N) is shown on the top.

Dust lidar ratio is larger for the Saharan Desert (Rg5) than the Arabian Desert (Rg6). The dust lidar ratio of the Saharan Desert measured by Raman lidar in Senegal during the SHADOW (study of SaHAran Dust Over West Africa) campaign shows 53 ± 8 sr at 532 nm [35], whereas Mamouri et al. [36] observed typical Middle Eastern dust lidar ratios around 40 sr from Raman lidar measurements in the Cyprus region. Note that, hereafter, all introduced lidar ratios are at the wavelength of 532 nm. Nisantzi et al. [23] found mean (± SD) lidar ratio of 53 ± 6 sr for Saharan dust and 41 ± 4 sr for Middle East dust at Limassol, Cyprus from polarization lidar measurements using the AERONET AOD as a constraint. Schuster et al. [11] retrieved dust lidar ratios from AERONET measurements and found median lidar ratio of 55.4 sr for the non-Sahel regions of Northern Africa and 42.6 sr for the Middle East. They also suggested that the relative proportion of the mineral illite is responsible for the larger dust lidar ratios in Western Sahara than Eastern Sahara and Arabian Desert. Shin et al. [37] reported similar results showing the lidar ratio is larger for Saharan dust than Arabian dust at different wavelengths of 440 nm and 675 nm using the AERONET inversion outputs. Córdoba-Jabonero et al. [38] compared lidar ratios for Saharan dust and Arabian dust retrieved by a synergetic use of the micro-pulse lidar and AERONET measurements. Their mean lidar ratios are 54 ± 5 sr for Saharan dust and 46 ± 2 sr for Arabian dust. The mean (± SD) lidar ratios retrieved in this study of 49.5 ± 36.8 sr for Saharan dust and 42.5 ± 26.2 sr for Arabian dust are consistent with previous studies. However, medians for the lidar ratio distributions of 41.7 sr for Saharan dust and 38.0 sr for Arabian dust are smaller.

Dust lidar ratios are relatively lower over ocean (Rg1–3) than land (Rg4–5). Dust aerosols in the Mediterranean Sea (Rg2) are transported mostly from Saharan Desert except for eastern part of the Mediterranean Sea where dust aerosols are frequently transported from the Middle East [23,39]. There are many studies on dust transported to the Mediterranean regions reporting similar lidar ratios with Saharan dust [22,40–44]. The mean (median) retrieved lidar ratio of 44.4 ± 15.9 (42.9) sr in Rg2 is comparable but lower than the lidar ratio for Saharan dust.

Dust lidar ratios over the Mid Atlantic (Rg1) and Arabian Sea (Rg3) are also lower compared to their source regions. The mean (median) lidar ratios are 40.3 ± 12.4 (39.3) sr for Rg1 and 37.5 ± 12.1 (36.4) sr for Rg3. Karyampudi et al. [45] suggests very low lidar ratio for Saharan dust over the Atlantic between ~25 sr and 35 sr from the Lidar In-space Technology Experiment (LITE) observation. Liu et

al. [46] reported a dust storm transported from western Africa to the Gulf of Mexico across the Atlantic Ocean from the CALIOP measurements. They found lidar ratios of 41–45 sr, which are similar but slightly larger than this study. Rittmeister et al. [47] also found a slightly larger lidar ratio of 43 ± 8 sr for Saharan dust over the tropical Atlantic from shipborne Raman lidar observations. Kanitz et al. [48] observed long-range transported Saharan dust plumes over the Atlantic with shipborne lidar and found that lidar ratios may decrease as they are transported to remote ocean because of aging effects. They measured mean lidar ratios of 64 and 50 sr for dust with a short residence time of 2–3 days but lower lidar ratios of around 45 sr for aged dust which is still larger than this study. There are many studies reporting larger lidar ratios (45–55 sr) for Saharan dust transported to Atlantic Ocean [49–52]. Similarly, a relatively low lidar ratio of 38 ± 5 sr was observed for Arabian dust transported to Maldives during the Indian Ocean Experiment (INDOEX) from a Raman lidar measurement [22].

This study found a mean (median) lidar ratio over the East Asian ocean (Rg4) of 41.8 ± 27.6 (37.1) sr. In spite of low median values, the retrieved dust lidar ratio is comparable with previous studies. Murayama et al. [53] observed lofted Asian dust in the free troposphere over Tokyo, Japan using a Raman lidar and found a lidar ratio of 43.1 ± 7.0 sr. Sakai et al. [54] measured dust lidar ratios of 46 ± 5 sr with a Raman lidar during an Asian dust event in Tsukuba, Japan. Chiang et al. [55] reported a mean dust lidar ratio of 44 ± 19 sr using simultaneous measurements of a lidar and a sun photometer in Taiwan. A similar lidar ratio of 46 ± 8 sr for coarse-dominated aerosols is obtained from airborne measurements of aerosol scattering and absorption near Korea and Japan [56]. Liu et al. [57] observed Asian dust in Japan with a high spectral resolution lidar (HSRL) and a Raman lidar and measured lidar ratios ranging from 42 to 55 sr, with a mean of 51 sr. Noh et al. [58] also reported lidar ratios of 51 ± 6 sr for Asian dust from Raman lidar measurements at Gwangju, Korea.

The lidar ratios for dust in Taklamakan and Gobi Desert area (Rg7) and Australia (Rg8) are significantly lower compared to other regions. The lidar ratios of 35 ± 5 sr [22] and 38 ± 7 sr [59] for dust from the Gobi Desert are observed in Beijing. These are lower than other regions and downwind side of East Asia (Rg4). Dust lidar ratio for Australia of 33 ± 3 sr at 675 nm reported by Shin et al. [37] are also lower than other regions. In this study, we found mean (median) lidar ratios of 35.0 ± 31.1 (27.8) sr and 35.4 ± 34.4 (25.5) sr for Rg7 and Rg8, respectively.

## 5. Summary

Dust lidar ratios are retrieved by a synergetic use of CALIOP and MODIS products for 5 years from 2007 to 2011. The CALIOP level 1 total attenuated backscatter data is used for the retrieval and the CALIOP level 2 aerosol profile product is used to determine dust layers. Quality assured (QA > 1 for DT ocean, QA = 3 for DB land) AOD data from the MODIS level 2 aerosol product is used as constraint. MODIS AOD retrievals and CALIOP attenuated backscatter profiles closer than 10 km from the center of MODIS pixel are defined as collocated measurements. Clouds are screened out for both CALIOP and MODIS. The retrieval is performed for the whole column of the atmosphere from 30 km to the surface adopting a constant lidar ratio of 30 sr for aerosols of clear air above the detected layers.

The retrieved dust lidar ratios show a log-normal distribution with mean (median) values of 39.5 ± 16.8 (38.1) sr and 46.6 ± 36.3 (39.2) sr for ocean and land, respectively. The mean values are comparable to the value of 44 sr currently used in the CALIOP level 2 aerosol algorithm [18] but the median values are relatively lower. There is a distinct regional variation in the retrieved dust lidar ratios. Dust lidar ratio is larger for the Saharan Desert (49.5 ± 36.8 sr) than the Arabian Desert (42.5 ± 26.2 sr), which is consistent with many previous studies. Dust aerosols transported to the Mediterranean Sea (44.4 ± 15.9 sr), Mid Atlantic (40.3 ± 12.4 sr), and Arabian Sea (37.5 ± 12.1 sr) show lower values compared with their source regions. An aging process of the long-range transported dust to remote ocean may be responsible for low lidar ratios. Dust lidar ratio over ocean in East Asia is 41.8 ± 27.6 sr is comparable with previous studies. Over Taklamakan and Gobi Deserts region the retrieved dust lidar ratios (35.5 ± 31.1 sr) show low values but still comparable with previous studies. Dust lidar ratios for Australia (35.4 ± 34.4 sr) are also relatively low compared with other regions.

Although the mean AOD difference between CALIOP and MODIS is small (close to zero), the distribution of the AOD difference shows that the CALIOP AOD is biased low. However, when including clear air AOD for CALIOP, AODs from the two sensors become more comparable. A conclusion that can be drawn from this is that retrieving only for the detected layers in the CALIOP algorithm is one of the major reasons for lower AODs for CALIOP than MODIS.

Lidar ratios retrieved in this study are strongly affected by MODIS AOD, because it is used as a constraint for the retrieval. Further investigation for an impact of the uncertainty in MODIS AOD to the retrieval may be helpful to improve the accuracy of the retrieved lidar ratio. Nevertheless, results of this study show that dust lidar ratio has a distinct regional variability and suggest that different dust lidar ratios should be applied for different regions in the CALIOP aerosol retrieval algorithm to reduce AOD uncertainty.

**Author Contributions:** Conceptualization and Methodology, A.H.O. and S.-W.K.; Formal analysis and validation, S.-W.K. and M.-H.K.; Writing—original draft, M.-H.K.; Writing—review & editing, A.H.O. and S.-W.K.; All authors have read and agreed to the published version of the manuscript.

**Funding:** This research was funded by the Korea Meteorological Administration Research and Development Program under Grant KMI2018-05010, and by the National Strategic Project-Fine Particle of the NRF funded by the Ministry of Science and ICT (MSIT), the Ministry of Environment (ME), and the Ministry of Health and Welfare (MOHW) (Grant No. NRF-2017M3D8A1092021).

**Acknowledgments:** The CALIPSO data were obtained from the NASA Langley Atmospheric Science Data Center (https://eosweb.larc.nasa.gov/project/calipso/calipso_table). The MODIS data were obtained through the Level-1 and Atmosphere Archive & Distribution System (LAADS) Distributed Active Archive Center (DAAC) at the NASA Goddard Space and Flight Center Data Center (https://ladsweb.modaps.eosdis.nasa.gov/).

**Conflicts of Interest:** The authors declare no conflict of interest.

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
