# Peer review of "Dust Lidar Ratios Retrieved from the CALIOP Measurements Using the MODIS AOD as a Constraint"

_remotesensing, doi:10.3390/rs12020251_

Round 1

Reviewer 1 Report

The CALIPSO, a space-borne satellite, was used to retrieve dust lidar ratio in this manuscript. MODIS AODs from the dark target algorithm over ocean and the deep blue algorithm over land are applied as a constraint. The author analyzed the global dust lidar ratio from 2007 to 2011 and found that there are 8 typical regions in the world, of which 7 are the major dust sources. Dust lidar ratio in the Taklimakan and the Gobi Desert region, and Australia region is relatively low, which is attributed to the lower AOD value of CALIPSO. The method of this manuscript is detailed, the algorithm is more advanced, and the research scale is large. However, there are several main problems as follows, and major revision is recommended.

Major comments:

CALIPSO has been operating steadily since its launch in 2006. Why is the study time scale only 2007-2011? The fourth generation of CALIPSO has been launched in 2016, and the resolution and error rate have been greatly improved compared to the previous three generations. It is recommended that the authors add data from recent years to better complement and confirm the reliability of the results. In the introduction section, a large number of contents describe the advantages and development process of CALIPSO. However, the necessity and significance of this study are not well introduced. MYD04_L2 10km AOD product is used as a constraint for the retrieval, but there is 3km MODIS AOD C6 product. The horizontal resolution of CALIPSO Level 2 aerosol products is 333 m. If 10 km data is used, the error is large. The CALIPSO profiles will increase a lot, which can guarantee the accuracy of the results.

4 The AOD of MODIS is different from that of CALIPSO. But is there certain comparability between the MODIS AOD and including clean air AOD for CALIPSO? The author needs to elaborate.

It is unreasonable for the author to use dust layer as the whole atmosphere column to retrieve the extinction coefficient. Firstly, as the author said, there are some missing values in Fig. 2c in the near surface, and no good retrieve, but there is still extinction coefficient in Fig. 2D. Secondly, there are different types of aerosols and aerosol subtypes at different altitudes. If only considered as dust aerosols, it will greatly affect the credibility of the results. The author needs to compare the extinction coefficient of CALIPSO Level 2 aerosol product with that of MODIS retrieval in this study. Why does the author need to retrieve when the vertical extinction coefficient is available? This study considers that the dust lidar ratio in Taklimakan Desert and Gobi region is relatively small, which is mainly caused by the smaller MODIS AOD. However, some studies have shown that the AOD of Taklimakan desert is not low at the time of dust events (Zong et al., 2015; Nan and Wang, 2018; Xie et al., 2019). There are not only deserts and Gobi in this region, but also many surface types. The author divides the region too large, which is also the main factor leading to the difference of the results of this study.

Reference:

Nan, Y., Wang, Y., 2018. De-coupling interannual variations of vertical dust extinction over the Taklimakan Desert during 2007–2016 using CALIOP. Science of the Total Environment, 633, 608-617.

Zong, X., Xia, X., Che, H., 2015. Validation of aerosol optical depth and climatology of aerosol vertical distribution in the Taklimakan Desert. Atmospheric Pollution Research, 6(2), 239-244.

Xie, G., Wang, M., Pan, J., Zhu, Y., 2019. Spatio-temporal variations and trends of MODIS C6. 1 Dark Target and Deep Blue merged aerosol optical depth over China during 2000–2017. Atmospheric Environment, 214, 116846.

Minor points:

Line 84 list describes the advantages and disadvantages of DB and DT retrieve AOD, and add references. Lines 63-78 and 80-90 CALIPSO and MODIS data sources or links need to be added. All the abbreviations need to be defined for the first time. Check through the manuscript. The author can combine two examples of Arabian Sea (Fig. 2) on October 18, 2009 and Taklimakan Desert (Fig. 8) on May 27, 2009, which not only shows the results of dust lidar ratio, but also shows that the dust lidar ratio in Taklimakan desert is relatively small, saving the length of the manuscript. There is a large sand lidar ratio in Central Europe. Why is there no division (Fig. 4)? The whole manuscript needs spaces before and after brackets, and the author needs to modify them. It is necessary to mark the legend in the Fig. 6. What do gray and black represent respectively? The author can't just use numbers to highlight the legend. The author considers a constant lidar ratio of 30 sr for aerosols of clear air above the detected layers to be inappropriate. Because different aerosol types and subtypes have different radar ratios, this will have a certain impact on the accuracy of the inversion results of dust lidar ratio.

Author Response

Comments and Suggestions for Authors

The CALIPSO, a space-borne satellite, was used to retrieve dust lidar ratio in this manuscript. MODIS AODs from the dark target algorithm over ocean and the deep blue algorithm over land are applied as a constraint. The author analyzed the global dust lidar ratio from 2007 to 2011 and found that there are 8 typical regions in the world, of which 7 are the major dust sources. Dust lidar ratio in the Taklimakan and the Gobi Desert region, and Australia region is relatively low, which is attributed to the lower AOD value of CALIPSO. The method of this manuscript is detailed, the algorithm is more advanced, and the research scale is large. However, there are several main problems as follows, and major revision is recommended.

Response: We appreciate your thoughtful and helpful comments and suggestions. We tried to answer your comments. Our ‘Reply’ is embedded below. All changes we have made in the revised manuscript are not only mentioned in each response for the reviewer’s comments, but can also be tracked using the “track changes” function. We hope we provide the appropriate answers.

Major comments:

CALIPSO has been operating steadily since its launch in 2006. Why is the study time scale only 2007-2011? The fourth generation of CALIPSO has been launched in 2016, and the resolution and error rate have been greatly improved compared to the previous three generations. It is recommended that the authors add data from recent years to better complement and confirm the reliability of the results.

Response: We used the most updated data for both CALIOP (version 4) and MODIS (collection 6). The algorithm version is not related with years of the data. The latest version (version 4) of the CALIOP algorithm are applied for whole priod of the data.

Longer period of the data would be better for the results. However, adding more years of data does not change any statistics of the retrieved lidar ratios. We think 5-year is enough for this study.

In the introduction section, a large number of contents describe the advantages and development process of CALIPSO. However, the necessity and significance of this study are not well introduced.

Response: CALIOP’s improvements in version 4 is introduced in the second paragraph in Introduction (line 34-45). The third and the fourth paragraph (line 46-61) introduce this study. We add additional sentence to emphasize the main purpose of this study.

MYD04_L2 10km AOD product is used as a constraint for the retrieval, but there is 3km MODIS AOD C6 product. The horizontal resolution of CALIPSO Level 2 aerosol products is 333 m. If 10 km data is used, the error is large. The CALIPSO profiles will increase a lot, which can guarantee the accuracy of the results.

Response: MYD04_3K (C6) product has 3km horizontal resolution. It can provide closer collocations than MYD04_L2 as the reviewer mentioned. However, the coloset approch is not always the best for the AOD comparison due to small scale inhomogeneoties (Schuster et al., 2012). We tried to use MYD04_3K (3x3 km2) with smaller collocation distance, but the results are almost unchanged. Sometimes closer collocation makes the results worse due to the retrieval failure.

The CALIOP level 1 profiles (333m) need to be averaged to retrieve aerosol extinction because of noisy signal. Although the CALIOP version 4 products provide a new data of merged cloud and aerosol “layer product” with horizontal resolution of 333m, aerosol “profile product” still has horizontal resolution of 5km. It is because horizontal averaging is needed to detect weak aerosol layers. To retrieve “clear air AOD”, we conclude we'd better to average the CALIOP level 1 proviles (333m) at least for 10 km. 

The AOD of MODIS is different from that of CALIPSO. But is there certain comparability between the MODIS AOD and including clean air AOD for CALIPSO? The author needs to elaborate.

Response: As mentioned in Introduction (line 32), retrieving only detected layers (not retrieving for undetected layers or clear air) is one of the main reasons for AOD discrepancy between CALIOP and MODIS. Figure 3 shows that, when including clear air AOD, the distribution of AOD difference becomes comparable. We re-organized the manuscript and Figure 6 moves to Figure 3.

It is unreasonable for the author to use dust layer as the whole atmosphere column to retrieve the extinction coefficient. Firstly, as the author said, there are some missing values in Fig. 2c in the near surface, and no good retrieve, but there is still extinction coefficient in Fig. 2D. Secondly, there are different types of aerosols and aerosol subtypes at different altitudes. If only considered as dust aerosols, it will greatly affect the credibility of the results.

Response: We retrieved only for dust layers, which means we did not retrieved when other types of aerosol exist. In other words, any collocated data pairs including other aerosol types are excluded from this study. Retrieving only dust layers for whole atmosphere does not mean that we treat whole atmosphere as dust. We separate detected dust layers from clear air and retrieve whole atmosphere from 30 km to the surface with different lidar ratios for clear air and dust layer.

Figure 2C is extinction coefficients from the CALIOP level 2 product which is not retrieved in this study. Since the CALIOP algorithm sometimes reports clear (or missing) space between the saperated layers, we ignored these empty space and considered as a single layer from the top of the highest layer to the surface, as shown in Figure 2D (which is retrieved in this study).

The author needs to compare the extinction coefficient of CALIPSO Level 2 aerosol product with that of MODIS retrieval in this study.

Response: MODIS provides only column integrated AODs, it does not provide vertically resolved profiles of aerosol extinction. We cannot compare the aerosol extinction coefficient from the CALIOP level 2 aerosol product with MODIS. But we compared CALIOP AOD (vertical integrated from the aerosol extinction coefficient) with MODIS AOD in Figure 3.

Why does the author need to retrieve when the vertical extinction coefficient is available?

Response: We do not need the vertical extinction coefficient (or AOD) from the CALIOP for the retrieval. We only need the CALIOP level 1 TAB profiles and MODIS AODs. However, we select collocated data pairs when the CALIOP level 2 aerosol extinction with QC is avaiable. It is because we need to compare AODs from the two sensors. The CALIOP AOD is a vertical integration of the extinction coefficients.

This study considers that the dust lidar ratio in Taklimakan Desert and Gobi region is relatively small, which is mainly caused by the smaller MODIS AOD. However, some studies have shown that the AOD of Taklimakan desert is not low at the time of dust events (Zong et al., 2015; Nan and Wang, 2018; Xie et al., 2019). There are not only deserts and Gobi in this region, but also many surface types. The author divides the region too large, which is also the main factor leading to the difference of the results of this study.

Response: We re-organized revised manuscript and Figure 7 and 8 are removed. In revised manuscript, we excluded data pairs with large AOD difference because large AOD difference implies that the AOD form either or both of two sensors may be uncertain. Therefore, discussion of AOD difference is less necessary and mostly removed from the paper.

We did not include or discuss about this comments because we only concerned about lidar ratios in the Taklamakan/Gobi Deserts.

Reference:

Nan, Y., Wang, Y., 2018. De-coupling interannual variations of vertical dust extinction over the Taklimakan Desert during 2007–2016 using CALIOP. Science of the Total Environment, 633, 608-617.

Zong, X., Xia, X., Che, H., 2015. Validation of aerosol optical depth and climatology of aerosol vertical distribution in the Taklimakan Desert. Atmospheric Pollution Research, 6(2), 239-244.

Xie, G., Wang, M., Pan, J., Zhu, Y., 2019. Spatio-temporal variations and trends of MODIS C6. 1 Dark Target and Deep Blue merged aerosol optical depth over China during 2000–2017. Atmospheric Environment, 214, 116846.

Minor points:

Line 84 list describes the advantages and disadvantages of DB and DT retrieve AOD, and add references.

Response: MODIS provides AODs with two algorithms of DT and DB. Over ocean, we have no choice but DT AODs. AODs are available for both DT and DB over land. However, we selected DB AODs over land because DB algorithm provides AODs over bright desert surface. We mentioned this in the text (line 83).

We think describing detailed advantages and disadvantages is out of our focus and less necessery. We introduced briefly about AODs of DT and DB in Section 2.2 and about their expected error in Section 2.3. We also introduced references for DT and DB MODIS AODs (Levy et al., 2013; Sayer et al., 2013).

Lines 63-78 and 80-90 CALIPSO and MODIS data sources or links need to be added.

Response: Data sources are introduced in “Acknowledgement”.

All the abbreviations need to be defined for the first time. Check through the manuscript.

Response: We checked again and improved. Thank you.

The author can combine two examples of Arabian Sea (Fig. 2) on October 18, 2009 and Taklimakan Desert (Fig. 8) on May 27, 2009, which not only shows the results of dust lidar ratio, but also shows that the dust lidar ratio in Taklimakan desert is relatively small, saving the length of the manuscript.

Response: We removed Figure 8.

There is a large sand lidar ratio in Central Europe. Why is there no division (Fig. 4)?

Response: In Figure 5, we analyze lidar ratios only for target regions; for major dust source regions over land and outflow regions over ocean (Rg1-8). There are lidar ratios in other regions such as Northern and Southern America, Europe and Central Asia. However, These regions are much interested compared to Rg1-8. The number of retrieved lidar ratio is smaller than target regions. Note that the color scale in Figure 5 is not a number of data.

The whole manuscript needs spaces before and after brackets, and the author needs to modify them.

Response: We check them again through the manuscript.

It is necessary to mark the legend in the Fig. 6. What do gray and black represent respectively? The author can't just use numbers to highlight the legend.

Response: We add legend for the gray and black bars.

The author considers a constant lidar ratio of 30 sr for aerosols of clear air above the detected layers to be inappropriate. Because different aerosol types and subtypes have different radar ratios, this will have a certain impact on the accuracy of the inversion results of dust lidar ratio.

Response: As the reviewer’s comments, the uncertainty of lidar ratio for clear air can be a source of error in the retrieval. Kim et al. (2017) investigate lidar ratio for clear air and found global mean value of ~30 sr. Because no aerosols are detected in clear air, we cannot classified aerosol types nor determine lidar ratios for clear air. There can be an error caused by clear air. However, it is not significant because clear air AOD is very low compared to AOD for detected aerosol layers. It is discussed by Kim et al. (2017).

Reviewer 2 Report

This study uses the synergy of CALIPSO and MODIS retrievals of total attenuated backscatter, extinction coefficient, AOD and vertical profiles in order to provide estimates of the dust Lidar Ratio (LR) over regions, both land and ocean, impacted by dust layers. LR values are provided for the main dust-laden regions around the world such as Sahara, Arabian Peninsula, Taklimakan-Gobbi deserts, east-Asian sea, Mediterranean Sea, Arabian Sea, subtropical Atlantic. The computed LRs from the synergy of CALIPSO and MODIS are compared with those from previous studies over these regions, mostly using lidar instruments and are in general agreement, except from the LRs over Taklimakan-Gobbi deserts and the desert of Australia. The reasons for that discrepancy are explained in detail via the comparison between CALIPSO and MODIS AODs. The paper is well organized and easy to follow, while some few parts need revision or better clarification. I attach a pdf file with specific comments to authors. However, the literature about LRs during dust-storm events is very rich, especially in the Mediterranean Basin, and can be also enriched in the manuscript.  

Author Response

Comments and Suggestions for Authors

This study uses the synergy of CALIPSO and MODIS retrievals of total attenuated backscatter, extinction coefficient, AOD and vertical profiles in order to provide estimates of the dust Lidar Ratio (LR) over regions, both land and ocean, impacted by dust layers. LR values are provided for the main dust-laden regions around the world such as Sahara, Arabian Peninsula, Taklimakan-Gobbi deserts, east-Asian sea, Mediterranean Sea, Arabian Sea, subtropical Atlantic. The computed LRs from the synergy of CALIPSO and MODIS are compared with those from previous studies over these regions, mostly using lidar instruments and are in general agreement, except from the LRs over Taklimakan-Gobbi deserts and the desert of Australia. The reasons for that discrepancy are explained in detail via the comparison between CALIPSO and MODIS AODs. The paper is well organized and easy to follow, while some few parts need revision or better clarification. I attach a pdf file with specific comments to authors. However, the literature about LRs during dust-storm events is very rich, especially in the Mediterranean Basin, and can be also enriched in the manuscript.

Response: We appreciate your thoughtful and helpful comments and suggestions. We tried to answer your comments. Our ‘Reply’ is embedded below. All changes we have made in the revised manuscript are not only mentioned in each response for the reviewer’s comments, but can also be tracked using the “track changes” function. We hope we provide the appropriate answers.

Comments on PDF:

Line 21: What;s about MODIS? Mention here if it overestimates or underestimates CALIOP-AOD and in what degree. This is a critical for the reliability of the LR values mentioned above.

Response: We re-organized the manuscript and removed data pairs with large AOD difference from the analysis. The sentence mentioning AOD difference in Abstract are removed.

Line 38-39: This sentence should be placed after the CALIOP discussions, in the next paragraph when you speak about MODIS. Also, you may refer in brief the MODIS uncertainties over land and ocean from the recent evaluation studies.

Response: It isn’t a comparison between AERONET and MODIS. It means CALIOP vs. AERONET and CALIOP vs. MODIS. The sentence is rephrased.

We introduced the expected errors for MODIS AOD reported by MODIS science team (Levy et al., 2013; Sayer et al., 2013). There are recent validation or comparison studies of the MODIS AODs. However, most of the validation studies are focused on limited area and limited cases. Because we used MODIS data globally for long period of time, we think it’s better to show general values of uncertainty.

Line 45: Are there several dust types in the aerosol classification by CALIPSO? I think you mean that fixed values are used for dust at any region over the globe. Please, check and rephrase the sentence.

Response: Correct! We modified the sentence.

Line 135-137: Why only clean air and undetected layers? What's about aerosols from other sources, i.e. anthropogenic, especially in areas surrounding the arid/desert regions in south Asia... Also, I think you mean clean not clear.. Please, clarify.

Response: We select only dust layers in this study. Any CALIOP profiles which contain non-dust (other types) aerosols are excluded. Although some other aerosols may be misclassified as dust, an effect of aerosols from other sources are negligible and not discussed in this study. CALIOP’s aerosol classification is the most accurate for dust among 7 aerosol types as mentioned in line 58-59. We modified some sentences and tried to make it clear (line 57 and line 100-101).

We used “clear” rather than “clean”. Clear air represents no aerosols are detected or no aerosols in there. Even for clear air, however, very weak (undetectable) aerosols can be exist. Thus, it need to be retrieved to be compared with MODIS AOD which is column integrated AOD. Clean air may represent “clean aerosols” from natural sources such as clean marine or clean continental aerosols.

Line 177-178: So, this should be termed as clean.

Response: We think that “clear” means nothing in there, but “clean” means something which is not contaminated. Because the CALIOP aerosol products uses terms of “clean marine” and “clean continental” as aerosol types, “clean air” can be confused as “clean aerosols”.

Line 202-204: I also think that extremely high values like 80-100 or even above are also unphysical for LR. Why may we have such very large values? Is is exclusively attributed to AOD retrievals, i.e. very high AOD over those pixels or even to very low TAB? More discussions are needed here.

Response: Lidar ratios can have any values in the retrieval even for unphysical values. Sometimes lidar ratios go to infinity or infinitesimal (close to zero) and the retrieval is failed. It is mainly due to error or random noise in the TABs. It is natural and we mentioned that 30~40% of retrievals are diverged in line 165 of the revised manuscript.

In the revised manuscript, we select collocated data pairs only for lower AOD difference (AOD difference lower than 1SD of the mean difference) to minimize unphysical lidar ratios. Also, we modified Figure 4 (omitting tails for both ends) to avoid misunderstanding. However, extreme (unphysical) lidar ratios are still exist. It is no meaning to discuss about these unphysical values. But they cannot be removed because mean and median values can be changed by arbitrary criteria for the lidar ratios.

Line 235: Has is also a linkage with the higher absorption of Saharan dust compared to Arabian dust? I think that this may be discussed.

Response: We introduced previous study discussing about this in line 270-272.

Line 253: the Middle East affects only the easternmost part of the Mediterranean and this should be mentioned here. So, the central and western Mediterranean are exclusively affected by Sahara.

Response: We modified whole paragraph.

Line 288: This is true, but the transported Taklimakan dust to Beijing is considered aged dust with generally lower LR.

Response: Thank you for the comment. Yes, it is possible. However, the references (Muller et al., 2007; Tesche et al., 2007) did not mention about aging effect for the low dust lidar ratio. Also, many studies reporting long-range transported dust to East Asia (East China, Korea, and Japan) do not show lower lidar ratio due to aging effect as mentioned in the text (line 301-311).

Line 311: It is not clear here, if CALIOP AOD includes only dust+clean air or even the anthropogenic AODs from organics, sulfate, nitrate, etc... This has to be clarified here.

Response: As mentioned above, we selected only dust aerosols and excluded all other types of aerosols.

Line 322-323: Revise as "... layer is very strong attenuated and is misclassified as cloud."

Line 323-325: For which layer? This sentence confuses the reader. Please rephrase and be more specific.

Line 330: Is there any explanation for the large discrepancy between MODIS and CALIPSO AODs over that region? It would be nice to discuss it in the manuscript, in order the reader the be convinced for large biased LR over Taklimakan and Gobbi deserts. Maybe MODIS DB retrievals are highly biased there, and this is the main reason for the large differences between the CALIPSO and MODIS AODs.

Response: We remove data pairs with large AOD difference between CALIOP and MODIS. Large AOD difference implies that either or both of the two sensors could be uncertain. Because collocated data pairs with large AOD difference are excluded, discussion about the AOD difference (Section 5) is removed from the revised manuscript.

Reviewer 3 Report

The authors present a very nice study of inferring the lidar ratio of mineral dust from a combination of spaceborne lidar and radiometer measurements. They make use of the latest data release of the CALIPSO retrieval so that their study also marks an update of any earlier study of that kind. The method and results are clearly presented. However, I believe that major revisions are necessary before this work should be accepted for publication.

Major comments:

We I read the paper I got the impression that the authors got the presentation backwards. I think it would be easier for a reader to follow the analysis if the AOD differences would be discussed before the retrieval and its output. In particular, I see it as a major flaw that the authors do not perform any screening with respect to the difference in AOD from MODIS and CALIPSO L2. Instead, they present Figure 8 and respective discussion to show that the method does not lead to meaningful findings when the AOD difference is high. It makes no sense to me to consider cases with AOD differences as high as 1.5! The authors should be more critical about this rather than presenting and discussion clearly too low dust lidar ratios. In fact, they should include a sensitivity study to estimate which differences are still acceptable for analysis. While including an AOD-difference constraint is going to decrease the number of useable retrievals (see page 5, lines 165/166), the results will be more reliable. As a consequence, Figure 8 and related discussion could be removed.

Minor comments:

There are plenty of studies with ground-based lidars and radiometers that employ coinciding AOD measurements for constraining the lidar retrieval. This should be clearly mentioned before the authors present that they have adapted this methodology for spaceborne measurements. The study is about lidar ratios but the definition of this parameter is given no earlier than page 4, line 151.  Figure 1: The figure presents a nice flow chart of the analysis. However, it uses symbols that have not been used in the text before and are also rarely used later, e.g. \tau instead of AOD. Also, the authors should clearly point out the connection between the extinction coefficient and AOD in the text. Also, is there any horizontal averaging in the retrieval as in the CALIPSO products? If not, it should be mentioned as another difference between the customized analysis and the standard product. page 4, line 153: is inform used correctly here? I am wondering why the authors limit the analysis to 5 years of data? Figure 3: Would it be possible to include the information on AOD differences in this plot? Maybe go to a 2d histogram? Figure 4: What is the size of the grid box? How many profiles are in the different boxes? I'd suggest to include an extra panel with number of profiles per grid box so that the readers can draw their own conclusions about the representativeness of the different values. Figure 5 and Table 1 basically show the same information. Please omit one. Do I understand correctly that clear air AOD is related to faint layers that are not detected by the CALIPSO feature finder? Maybe this should be stated explicitly somewhere?

Author Response

Comments and Suggestions for Authors

The authors present a very nice study of inferring the lidar ratio of mineral dust from a combination of spaceborne lidar and radiometer measurements. They make use of the latest data release of the CALIPSO retrieval so that their study also marks an update of any earlier study of that kind. The method and results are clearly presented. However, I believe that major revisions are necessary before this work should be accepted for publication.

Response: We appreciate your thoughtful and helpful comments and suggestions. We tried to answer your comments. Our ‘Reply’ is embedded below. All changes we have made in the revised manuscript are not only mentioned in each response for the reviewer’s comments, but can also be tracked using the “track changes” function. We hope we provide the appropriate answers.

Major comments:

We I read the paper I got the impression that the authors got the presentation backwards. I think it would be easier for a reader to follow the analysis if the AOD differences would be discussed before the retrieval and its output. In particular, I see it as a major flaw that the authors do not perform any screening with respect to the difference in AOD from MODIS and CALIPSO L2. Instead, they present Figure 8 and respective discussion to show that the method does not lead to meaningful findings when the AOD difference is high. It makes no sense to me to consider cases with AOD differences as high as 1.5! The authors should be more critical about this rather than presenting and discussion clearly too low dust lidar ratios. In fact, they should include a sensitivity study to estimate which differences are still acceptable for analysis. While including an AOD-difference constraint is going to decrease the number of useable retrievals (see page 5, lines 165/166), the results will be more reliable. As a consequence, Figure 8 and related discussion could be removed.

Response: Thank you for your valuable comment. We re-organized whole manuscript as the reviewer’s comment. We selected collocated data pairs with AOD difference smaller than ±1SD. By accepting relatively strict criteria for data selection, we don’t need to discuss about the AOD difference (Section 5). We also removed Figure 7 and 8.

Minor comments:

There are plenty of studies with ground-based lidars and radiometers that employ coinciding AOD measurements for constraining the lidar retrieval. This should be clearly mentioned before the authors present that they have adapted this methodology for spaceborne measurements.

Response: We mentioned this in Introduction with four references (line 46-47).

The study is about lidar ratios but the definition of this parameter is given no earlier than page 4, line 151.

Response: We add some definition and importance of the lidar ratio in Introduction (line 32 and 38-39).

Figure 1: The figure presents a nice flow chart of the analysis. However, it uses symbols that have not been used in the text before and are also rarely used later, e.g. \tau instead of AOD.

Response: We checked all parameters are introduced in the text (mostly equations 1 and 2). Epsilon is not introduced in the text but defined in the flowchart. Tau_MODIS and Tau_CALIOP is used only in the flowchart for the convenience. It is not appropriate to use “MODIS AOD” or “CALIOP AOD” instead of tau for the equations in Figure 1. But in the text below the Figure 1, we think “AOD” (MODIS AOD and CALIOP AOD) is easier to recognize than tau.

Also, the authors should clearly point out the connection between the extinction coefficient and AOD in the text.

Response: In line 160, in the text just below Figure 1, we mentioned that AOD is obtained from vertical integration of the retrieved aerosol extinction. Also, equation for AOD is shown in Figure 1.

Also, is there any horizontal averaging in the retrieval as in the CALIPSO products? If not, it should be mentioned as another difference between the customized analysis and the standard product.

Response: In this study, we average the CALIOP level 1 TAB profiles for each collocated data pair. We add a sentence for this in line 98-99.

page 4, line 153: is inform used correctly here?

Response: The sentence is rephrased.

I am wondering why the authors limit the analysis to 5 years of data?

Response: Longer period of the data would be better for the results. However, adding more years of data does not change any statistics of the retrieved lidar ratios. We think 5-year is enough for this study.

Figure 3: Would it be possible to include the information on AOD differences in this plot? Maybe go to a 2d histogram?

Response: As the reviewer’s comment, AOD difference and the retrieved lidar ratio could show some correlation. However, after removing collocated data pairs with large AOD difference, plotting AOD difference together in Figure 3 (Figure 4 in the revised manuscript) is less meaningful.

Figure 4: What is the size of the grid box? How many profiles are in the different boxes? I'd suggest to include an extra panel with number of profiles per grid box so that the readers can draw their own conclusions about the representativeness of the different values.

Response: We think that showing number of retrieved data with additional map is less necessary. Number of data for each region (white boxes in Figure 5) is shown in Figure 5.

Figure 5 and Table 1 basically show the same information. Please omit one.

Response: We remove Table 1 but showed the number of data on the top of Figure 6.

Do I understand correctly that clear air AOD is related to faint layers that are not detected by the CALIPSO feature finder? Maybe this should be stated explicitly somewhere?

Response: We add a definition for clear air in line 137.

Round 2

Reviewer 1 Report

The author has made a number of substantial changes to the manuscript. This makes the manuscript structure more organized and the logic clearer. However, some parts need further modification to make the paper more complete. So I recommend reconsideration of your paper following minor revision

1) The author says that five years is enough to carry out this research, but the manuscript year is 2007-2011, 8 years have passed, and the author needs to supplement the latest data to verify the authenticity and reliability of this work.

2) The introduction part needs to be re-written again. It is not enough to add some sentences. The overall logic is not rigorous enough. It does not lead to this research work.

3) The small lidar ratios in the Taklamakan / Gobi Deserts may be due to the large range of China. The Gobi and deserts in northwest China are widely distributed, but the degree of urbanization in eastern China is very high. The range should not be divided. It's too big.

The author will consider accepting the manuscript when the above issues are resolved.

Author Response

Dear Editor and reviewer,

We appreciate your thoughtful and helpful comments and suggestions. We tried to answer your comments. Our ‘Reply’ is embedded below. We hope we provide the appropriate answers. If you need any further comments, please let us know.

Best regards,

Authors

The author has made a number of substantial changes to the manuscript. This makes the manuscript structure more organized and the logic clearer. However, some parts need further modification to make the paper more complete. So I recommend reconsideration of your paper following minor revision

1) The author says that five years is enough to carry out this research, but the manuscript year is 2007-2011, 8 years have passed, and the author needs to supplement the latest data to verify the authenticity and reliability of this work.

Response: Thank you again for taking your time to read and give comments. Basically, it has been reported that the dust lidar ratio for a certain region does not show inter-annual variation. Based on your comments in the first review, we calculated the lidar ratio for dust particles for the whole year of 2018, and found there was no difference in dust lidar ratio compared with the results over the period of 2007-2011. On the other hand, as we noted in the manuscript, different dust lidar ratios for different dust source regions are great concern in CALIPSO and lidar communities. We think, therefore, that 5-year CALIPSO and MODIS data are enough to discuss the lidar ratios for different dust source regions.

2) The introduction part needs to be re-written again. It is not enough to add some sentences. The overall logic is not rigorous enough. It does not lead to this research work.

Response: The Introduction has four paragraphs to emphasize why we revisit the lidar ratio of dust particles over different dust source regions and to explain the advantage of CALIOP-MODIS synergy. In the first paragraph, we introduces lower CALIOP AODs due mainly to lidar ratio. The second paragraph explained recent improvements in the CALIOP’s aerosol algorithm, which still has some uncertainties in lidar ratio. AOD constrained retrieval method was described in third paragraph. The last paragraph briefly describes overall goals of this study. We have already modified the Introduction section in last revision, and decide to keep it unchanged.

3) The small lidar ratios in the Taklamakan / Gobi Deserts may be due to the large range of China. The Gobi and deserts in northwest China are widely distributed, but the degree of urbanization in eastern China is very high. The range should not be divided. It's too big.

Response: In the first revised manuscript (Round 1), area for Rg7 (Taklamkan/Gobi) was rearranged and urban areas of the Eastern China were excluded to minimize the effects of urban pollution aerosols on dust lidar estimation.

Thank you again for your comments.

-End of Document-

Reviewer 3 Report

The authors have addressed my concerns about this work appropriately. I have no further comments.

Author Response

Thank you for the valuable comments.